# Investigation of the Microstructure, Hardness and Corrosion Resistance of a New 58Ag24Pd11Cu2Au2Zn1.5In1.5Sn Dental Alloy

**DOI:** 10.3390/ma12244199

**Published:** 2019-12-13

**Authors:** Dan Gheorghe, Ion Pencea, Iulian Vasile Antoniac, Ramona-Nicoleta Turcu

**Affiliations:** Materials Science and Engineering Faculty, University Politehnica of Bucharest, 313 Splaiul Independentei, PC-060042 Bucharest, Romania; ghe58dan@yahoo.com (D.G.); antoniac.iulian@gmail.com (I.V.A.); ramona.nicoleta.turcu@gmail.com (R.-N.T.)

**Keywords:** 58Ag24Pd11Cu2Au2Zn1.5In1.5Sn dental alloy, elemental analysis, structural analysis, Vickers hardness tests, corrosion test

## Abstract

Higher-noble dental alloys (Au, Ag, and Pd) are the most desirable for dentistry applications, but they are expensive. Low-noble (Ag, Pd, Cu) dental alloys are alternatives to higher-noble ones due to their lower price. In this regard, the paper supports the price lowering of dental alloy by increasing the Cu content, i.e., a new 58Ag24Pd11Cu2Au2Zn1.5In1.5Sn dental alloy. The increasing addition of the Cu leads to a complex structure consisting of a solid solution that engulfs compounds of micrometric and nanometric sizes. The 58Ag24Pd11Cu2Au2Zn1.5In1.5Sn has demonstrated a much better electrochemical corrosion behavior in artificial saliva compared to the Paliag and Unique White dental alloys. The improved corrosion behavior of the new alloy is supported by the diminishing of the Cu selective diffusion into the electrolyte due to its retaining into compounds and into Ag-Pd solid solution. Also, the synergic effects of Cu, Zn, In, Sn may improve the corrosion resistance, but they have strengthened the 58Ag24Pd11Cu2Au2Zn1.5In1.5Sn matrix. The main finding addressed in the paper consists in a new 58Ag24Pd11Cu2Au2Zn1.5In1.5Sn dental alloy with improved corrosion resistance in artificial saliva.

## 1. Introduction

On the market, there is a wide variety of dental alloys, ranging from nearly pure gold and conventional gold-based alloys to alloys based on silver, palladium, nickel, cobalt, iron, titanium, tin, and other metals [1,2,3,4,5]. The best choice in this area is a gold alloy due to its high mechanical properties, good corrosion resistance, and excellent biocompatibility, but its price still impedes this choice. Accordingly, alternative materials such as Ag-Pd alloys, Co-Cr alloys, and Ti alloys have been introduced in dentistry practice. Recent trends have shown that low-noble dental alloys (Ag and Pd alloys, known as white alloys) are attracting much attention as alternatives to Au alloys because of their lower price [6,7,8]. Thus, Ag-based alloys such as Ag-Pd-Cu-Au [9,10,11,12,13,14,15,16,17,18,19], and Ag–Cu–Pd–Au [20,21] alloys have reached commercial status. Alloys like Unique White 66Ag 22Pd 10Cu 1In (wt%) and Paliag 50Ag 30Pd 15 9Cu 3Au 1Zn have achieved large recognition on the dental alloy market [22]. One of the way to further lowering the cost of the Ag-Pd-Cu-Au alloys is to lower the Pd content and to increase the Cu one as it was reported by different authors: Ag-20.9Pd-8.5Cu-12.5Au-2.5Zn-0.5Sn-0.1Ir [23]; Ag-20Pd-20Cu-12Au-2(Zn, Ir and In) [24]; Ag-25.4Pd-21.8Cu-10Au [25]; Ag-25.20Pd-9.88Cu-20Au, Ag-30.32Pd-9.66Cu-5.04Au; Ag-25.4Pd-12.82Cu-9.96Au; Ag-28Pd-9.12Cu-12.04Au [23], etc. Thus, the dental casting of Ag–20Pd–14.5Cu-12Au alloy (wt%) is widely used in Japan [6]. The Ag-Pd alloy system has the great advantage that Ag and Pd have complete miscibility in all composition ratios. The addition of Cu leads to the hardening of as-cast alloys and to improving the mechanical properties of the matrix. Also, Cu is believed to bring a bactericide behavior of the dental implant. The hardening mechanism consists in the formation of a CuPd ordered phase (L20-type) [26], precipitation of a Cu rich α1 phase [27], and phase separation of α solid solution to a CuPd ordered phase (L20-type) and Ag-rich α2 phase formation [7]. The addition of Au to Ag-Pd alloy aims the corrosion resistance increasing. The In addition improves the color change resistance of the alloy [18]. The addition of Zn aims the improving the castability of the alloy by entrapping the dissolved oxygen into the melt. The dissolved oxygen is violently released during Ag solid solution solidification and cause casting defects. The addition Sn to the Ag-Pd-Cu matrix aims the increase the mechanical strength by refining the micro-structure through micronized intermetallic compounds. Also, the addition of Zn, Sn, and In increase the adhesion of the dental enamels to the Ag-Pd-Cu crowns by chemical mechanism, i.e., chemical bonds among Zn, Sn, In that have diffused onto the metallic surface and the oxygen provided by the dental enamel. In this regard, the paper addresses a new alloy designed to improve the price-properties compromise. The envisaged new alloy is designed for the crown body that will be subsequently coated with porcelain enamel. The increased quantity of Cu into the matrix may lead to an increased risk of Cu leakage into surrounding tissues as it was reported [27]. The leakage of Cu, but the leakage of Zn, Sn, or In are prevented by the enamel coating of the crown or bridges made of this new alloy. Also, the usage of the new alloy as a substrate for the enameled crown eliminates any requirement for aging or other thermal treatment.

To ensure that the 58Ag24Pd11Cu2Au2Zn1.5In1.5Sn is fitted to the purpose it was considered firstly to assess its corrosion behavior in artificial saliva and to compare corrosion behavior to the similar dental alloys as Unique White and Paliag. Also, the corrosion behavior of the new dental alloy was supported by structural investigations carried with qualitative and quantitative optical microscopy (QOM), scanning electron microscopy (SEM) equipped with X-ray energy dispersion spectrometer (EDS), X-ray diffraction (XRD), and X-ray fluorescence spectroscopy (XRFS). Accordingly, structural characterization, inclusionary state assessment, elemental analysis, and hardness measurements provided quite enough information to assess the adequacy of the alloy to the intended usage. On the other hand, the complex characterization of the new alloy highlighted the need for further researches aimed to improve its functional properties (corrosion resistance, mechanical strength, Young modulus, and the adhesion of enamel to this new alloy).

## 2. Materials and Methods

The 58Ag24Pd11Cu2Au2Zn1.5In1.5Sn alloy was produced by induction melting in a vacuum from the high-purity precursors and poured into Cu molds as ingots with the diameter of 8 mm and 10 mm length. The bulk chemical composition was measured by XRFS using a XEPOS instrument (Spectro Xepos, Ametek De, Berwyn, PA, USA). The elements concentration (wt%) were calculated with the Turboquant-alloy analytical software version 01, SpectroDe, Germany) based on the intensities of the characteristic lines of the elements into specimens. The specimens for XRFS measurement were prepared by re-melting the ingots and powered into Cu mold of 32 mm in diameter to obtain disks of about 1 mm thick. The specimens were measured three times in repetitive conditions for a better estimation of the expanded measurement uncertainty with a 95% confidence level.

The microstructure and the inclusionary states of the new alloy were observed using a Reichert UnivaR optical microscope (Reichert AG, Wien, Austria) which is computer-assisted for image acquisition and processing by a Buehler Omnimet Enterprise software (version 4.2., Buehler, Lake Bluff, IL, USA). The specimens were cross-sectioned using a low-speed Buehler (Lake Bluff, IL, USA) saw under wet cooling jet to minimize the structure degradation. Subsequently, the cross-sections of the specimens were polished in two steps using a “Buehler-MICROFLOCK” textile backing and “Buehler-METADI” diamond powder suspensions with particle diameters between 1 µm and 15 µm. The finely polished cross-sections were subjected to chemical attacks using royal water reactive (60 mL HCl, 20 mL HNO_3_). The polished surfaces of the specimens were required mainly for optical microscopy investigations, but they were used also for SEM, XRFS, and XRD measurements.

The SEM-EDS investigations were carried out with a QUANTA INSPECT F 50 microscope (FEI-Philips, Eindhoven, The Netherlands) equipped with an EDS analyzer having an MnKα line resolution of 133 eV. Two purposes were pursued by the SEM-EDS measurement, i.e., a better morphology observation at higher magnification, but elemental distribution at micron-scale to assess the way in which the alloying elements are distributed into the matrix.

A PANalytical diffractometer type X’Pert PRO MPD (Instruments, Almelo, The Netherlands) with a Bragg–Brentano geometry was used for the qualitative and quantitative phase analyses of the specimens. The Cu Kα radiation emitted by the X-ray tube, operated at HV = 40 kV, Ia = 35 mA, was monochromatic before the entrance into the detector. The 2θ scan was performed with 0.02°/step with 2 s dwell time.

The Vickers Hardness was measured with a Hanemann hardness tester (Carl Zeiss, Jena, Germany).

Corrosion resistance tests were carried with a Potentiostat/Galvanostat (model PARSTAT 4000, Princeton Applied Research, Ametek, Berwyn, PA, USA) and the potentiodynamic curves (Tafel curves) were acquired using the VersaStudio software (version 02, Princeton Applied Research—AMETEK, Oak Ridge, TN, USA). Linear polarization technique was applied according to ASTM G5–14e1 [28]. Fusayama-Meyer artificial saliva was used for corrosion tests (chemical composition: 0.4 g·L^−1^ NaCl, 0.9 g·L^−1^ KCl, 1 g·L^−1^ urea, 0.69 g·L^−1^ NaH_2_PO_4_, 0.795 g·L^−1^ CaCl * 2H_2_O; pH = 5.2). The electrochemical tests were performed at human body temperature (37 ± 0.5 °C) using a Jeio Tech thermostatic bath (Jeiotech, Daejeon, Korea), model CW-05G.

## 3. Results and Discussion

The corrosion behavior is defining for a dental alloy. We chose to present first the results for alloy composition, microstructure, phase contents, and hardness for the reader to have the information important for understanding the corrosion test results.

### 3.1. Elemental Analysis

The averaged elemental concentrations (wt%) of the new alloy, measured with XEPOS (Spectro Xepos, Ametek De, Berwyn, PA, USA), are presented in Table 1. For a better estimation of the expanded uncertainty with a 95% confidence level (U (95%)), three specimens were measured three times in repetitive conditions. The U (95%) was estimated based on a proper top-down method which is described elsewhere [24].

As shown in Table 1, the melting process and subsequently the heat treatment applied to the specimens slightly modified the designed chemical composition of the alloy i.e., Zn, In, and Sn had suffered a kind of loss on ignition and, as a matter of consequence, Pd and Cu had increased their concentrations.

### 3.2. Optical Microscopy Analysis

The representative microstructure of the as-cast specimens is shown in Figure 1 at different magnifications. Noteworthy that the structure is observed following a chemical attack with royal water reactive (60 mL HCl, 20 mL HNO_3_).

The number of solid solutions and compounds was estimated by image processing with Omnimet Enterprise software as shown in Figure 2 and Figure 3.

The analysis of Figure 2 shows that the new alloy consists of about 68–70% eutectic, 24–26% solid solution and 1–2% compounds. The microstructure shown in Figure 3 looks like that in Figure 2, but it shows about 2–2.5% voids or pores.

The inclusionary state of the area in Figure 1a is shown in Figure 4.

The inclusionary state of the area in Figure 2a is shown in Figure 5.

### 3.3. SEM–EDS Analysis

The optical analysis has shown a complex pattern of the new alloy microstructure due to a complex mixture of eutectic, solid solution and compounds whose natures are expected to differ as Cu, Pd, Zn, In, and Sn can combine given rise to different compounds. The SEM–EDS are the best fitted for revealing the complex pattern of matrix and compounds and to provide quantitative information about compound natures by EDS elemental analysis as is very well depicted in Figure 6 and Figure 7. The images in Figure 6 were taken on an area considered as eutectic. At lower magnification, the area shows a uniform like morphology (Figure 6a) while at greater magnification (Figure 6b,c) the morphology is heterogeneous.

The pattern shown in Figure 6a–c is quite the same for the entire observed surface as is supported by Figure 7 taken on another field of eutectic.

Figure 7 provides a closer look at the microstructure of the as-cast 58Ag24Pd11Cu2Au2Zn1.5In1.5Sn alloy. On the red marked zones in Figure 7, EDS and dimensional measurements were carried on to assess the nature and the extent of the structural heterogeneities. Thus, the needle-like darker entities of about 100 nm in diameter (Figure 7b) include smaller entities of about 50 nm. Onto some areas, the needle-like entities are aligned parallel each-one with an average distance between rods of about 1 µm. The increased content of Cu was expected to give rise to a significant chemical heterogeneity which is supported by the microstructural patterns given in Figure 6 and Figure 7. The EDS measurements on the spotted areas in Figure 7a have confirmed the envisaged heterogeneity as is given in Table 2.

The concentrations of Ag and Pd (wt%, at%) do not vary significantly from one zone to another, but those of In, Sn, Cu, Zn, and Au vary significantly. Thus, into zone 1 Pd, Cu, and Zn show greater concentration while In and Sn are below the limit of detection. These aspects can be explained if the area is a Pd-Cu-Zn compound of the form PdCuZn_0.2_. Cu concentration decreases into zone 2, Zn is missing while Sn, In, and Au increases indicating an Sn-In-Au compound. Zone 3 is enriched in Cu and Zn while Au is missing and In, Sn slightly decrease. Thus, zone 3 may consist of a Cu-Zn compound.

A more relevant perception of the chemical heterogeneity is given by the EDS maps of element distributions as is shown in Figure 8 which depicts the distribution of the alloying elements associated with Figure 7a.

### 3.4. XRD Analysis

Qualitative phase analysis was performed based on ICDD data files using EVA search-match software. As could be seen in Figure 9 it was identified the solid solution of Ag-Pd-Cu having an Fm3m crystalline structure with a modified parameter of 0.0403 nm compared to 0.0409 nm of the pure Ag.

The inset pie chart in Figure 9 reveals the phase contents estimated by the Rietvelt method. The identified compounds are Ag_9_In_4_, Cu_0.7_Zn_2_, Cu_4_Pd, In_0.75_Sn_0.25_. The TOPAS software (version 1.0, PANanalytical B.V., Almelo, The Netherlands was used to estimate the phase content based on the diffractogram given in Figure 9.

The SEM-EDS data lets one expects the occurrence of ternary or even quaternary compounds into a new alloy, but XRD data supports the incidence of only binary compounds. At first glance, the phase content estimated based on the TOPAS Rietvelt method seems to agree with the SEM observations.

### 3.5. Vickers Hardness Test

The traces of the Vickers indentations on three representative areas of a polished specimen of 58Ag24Pd11Cu2Au2Zn1.5In1.5Sn are shown in Figure 10. The average microhardness of the specimen is 206 µHV40 which is significantly higher than that of the Paliag whose value is 180 µHV40 [19].

The higher microhardness of the 58Ag24Pd11Cu2Au2Zn1.5In1.5Sn could be assigned to the Cu addition and to the increased content of micron and nano-sized compounds whose occurrence can be assigned to Cu addition as well.

### 3.6. Corrosion Behaviour

#### 3.6.1. Open Circuit Potential

Knowledge of the corrosion behavior of the new alloys is essential to understanding its biocompatibility behavior. In this regard, the potentiostatic/potentiodynamic polarization test was considered as fit to purpose. The open circuit potential value (E_OC_) depends on the nature of the metal–solution interface, but mainly on the free Gibbs enthalpy of each element into electrode and into electrolyte. The E_OC_ of 58Ag24Pd11Cu2Au2Zn1.5In1.5Sn alloy varies as a function of time (Figure 11) but it quite stabilizes around 90 mV value after 7 h.

The downward spikes of the Eoc graph in Figure 11 can be assigned to the inclusionary state of the as-cast alloy. An inclusion can be the place of a fast cathode reaction that leads to a fast decreasing of E_OC_ followed by revenue caused by the self-healing of the protective layer.

#### 3.6.2. Potentiodynamic Polarization Curves

The electric potential vs. intensity (E–I) plots of the new dental alloys allow further insight into the corrosion mechanism. The Tafel plots (Figure 12) were obtained by applying an increasing electric potential on the working electrode with a rate of 0.167 mV/s in the range [Eoc−250 mV, Eoc+250 mV].

The Tafel test was carried on after 7 h exposure into artificial saliva at 37.5 °C. The zero-current potential (E_cor_) and corrosion current (Icor) were calculated using the standard methods based on acquired potentiodynamic polarization plots. The Tafel slopes (b_a_ and b_c_) were determined by fitting the ending parts of the polarization curves to lines. The two Tafel lines intercept at the point of the coordinates (Ecor, Icor). As was reported [19], an alloy with a tendency toward passivation will have a value of b_a_ > b_c_, while an alloy that corrodes will have a b_a_ < b_c_. The Icor is the most representative for the degradation degree of the alloy as it measures the net ion flux density through the double electric layer around the working electrode.

The polarization resistance of the alloy was estimated according to ASTM G59-97(2014) [29] using the equation:(1)$$R_{p}=\frac{1}{2.3}\frac{b_{a}|b_{c}|}{b_{a}+\left|bc\right|}\frac{1}{I_{cor}}.$$

The significances of the terms in Equation (1) have already specified above.

The corrosion rate was calculated according to ASTM G102-89 (2015)e1 [30] using the formula:(2)$$CR=K_{i}\frac{I_{corr}}{\rho }EW,$$
where CR—corrosion rate, K_i_ = 3.27 × 10^−3^—experimental constant, ρ—density of the working electrode, EW—equivalent weight

The main results of the electrochemical corrosion process are presented in Table 3.

The 58Ag24Pd11Cu2Au2Zn1.5In1.5Sn corrosion behavior is surprising comparing to the most similar dental alloy as Paliag and Unique White, manly from the E_OC_ point of view, i.e., the new alloy is prone to passivation (E_OC_ = +90 mV) while Paliag and Unique White are prone to corrosion as negative values were reported for them, i.e., −75 mV and −25 mV respectively [19]. Also, the value of the b_a_ obtained in our experiments (ba = 380 mV/nA) is two times greater than those reported for Paliag and Unique White (180 mV/nA, 185 mV/nA respectively) [19]. The 58Ag24Pd11Cu2Au2Zn1.5In1.5Sn (−108 mV/nA) b_c_ value can be considered similar to those reported for Paliag and Unique White (−135 mV/nA and −145 mV/nA), but still higher indicating a less tendency to corrosion than Paliag and Unique White [19].

The positive E_OC_ can be explained by the dominant contribution of the many compounds types that occur into the specimen to the corrosion mechanisms as it is shown in Figure 13.

The studied alloy consists mainly of a eutectic (brown areas in Figure 1, Figure 2 and Figure 3) that engulfed Ag-Cu, Cu-Zn, Ag-Cu, Ag-In, In-Sn binary compounds and, the most probably, ternary ones of the form Pd-Cu-Zn (Figure 8). These compounds nucleate and growth before eutectic solidification of the matrix. If the primary formed compounds have lower Gibbs free enthalpy than that of the eutectic, and then they are prone to oxidize and to positively charge the working electrode. Whatever the corrosion mechanism of the new alloy, the results of the electrochemical corrosion test carried on it prove its better corrosion behavior in artificial saliva rein comparison to the similar reported data for Paliag and Unique White. Another hypothesis that would be worth investigating consists of hindering the Cu selective diffusion into the electrolyte due to its retaining into compounds and into the Ag-Pd matrix. Also, the higher hardness of the new alloy can be assigned to the Cu hardening effect through the medium of compounds and matrix strengthening.

## 4. Conclusions

The approach to lower the price of the Ag-Pd base dental alloy by increasing the Cu content is effective as the 58Ag24Pd11Cu2Au2Zn1.5In1.5Sn has adequate mechanical properties.

The addition of Cu in the Ag-Pd based alloy resulted in a complex structure consisting of a solid solution that engulfs compounds of micrometric and nanometric sizes.

The element distributions into eutectic are heterogeneous.

The structural investigation confirms the higher value of hardness of the new dental alloy 58Ag24Pd11Cu2Au2Zn1.5In1.5Sn compared to commercial dental alloys type Paliag and Unique White.

Microstructural investigations provided valuable data for explaining the 58Ag24Pd11Cu2Au2Zn1.5In1.5Sn alloy electrochemical corrosion behavior, and its adequacy usage for dental crown and bridges that will be coated with porcelain enamel. Also, it was observed that the corrosion behavior of new dental alloy 58Ag24Pd11Cu2Au2Zn1.5In1.5Sn is better than that of commercial dental alloys type Paliag and Unique White. Future research will be made on the samples coated with porcelain enamel, in order to show the clinical benefit for metal-ceramic applications in dentistry.

## Figures and Tables

**Figure 1 materials-12-04199-f001:**
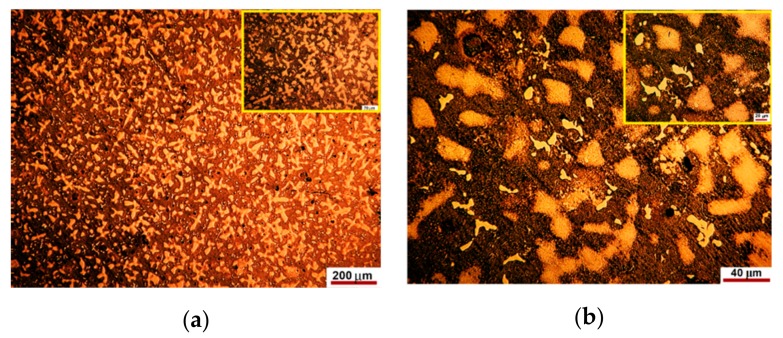
The representative microstructure of the as-cast specimens revealed at different magnifications. (**a**,**b**) typical casting microstructures that highlight solid dendrites based on Ag-Pd solid solution. The closer view in the upper right corners reveals structure aspects at higher magnification.

**Figure 2 materials-12-04199-f002:**
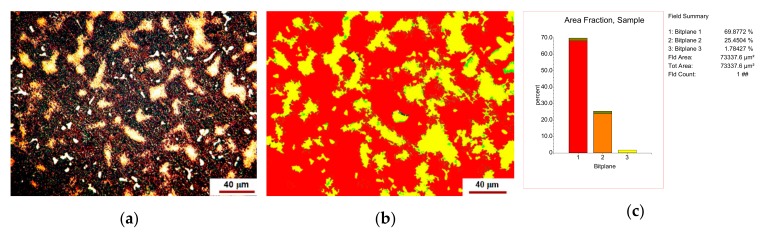
Phase content of the new alloy: (**a**) original microstructure image; (**b**) processed image; (**c**) histogram of the phase contents (% area), i.e., 1—eutectic, 2—solid solution, 3—compounds-yellow shinning ones.

**Figure 3 materials-12-04199-f003:**
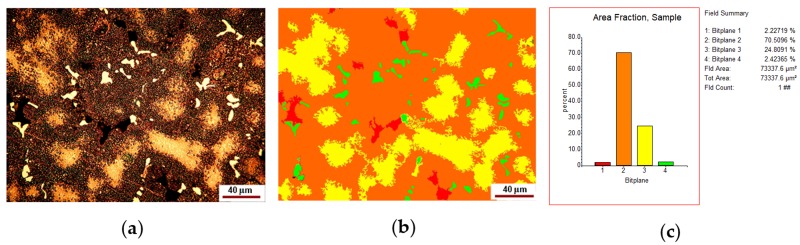
Phase content into other area: (**a**) original microstructure image; (**b**) processed image; (**c**) histogram of the phase contents (% area), i.e., 1—compounds, 2—eutectic, 3—solid solution, 4—voids or pores.

**Figure 4 materials-12-04199-f004:**
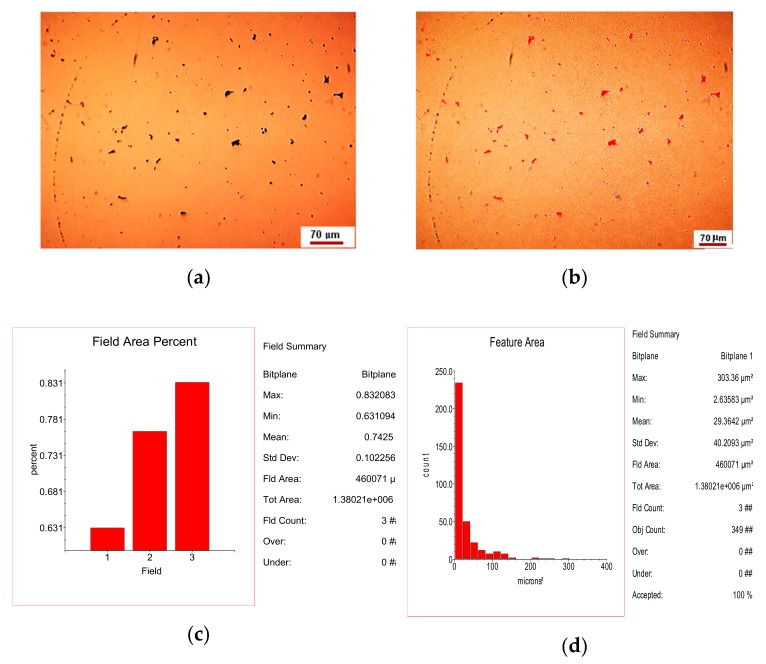
Inclusionary state associated with Figure 1a: (**a**) original microstructure image; (**b**) processed image; (**c**) histogram of the phase distribution; (**d**) histogram of the inclusion size distribution.

**Figure 5 materials-12-04199-f005:**
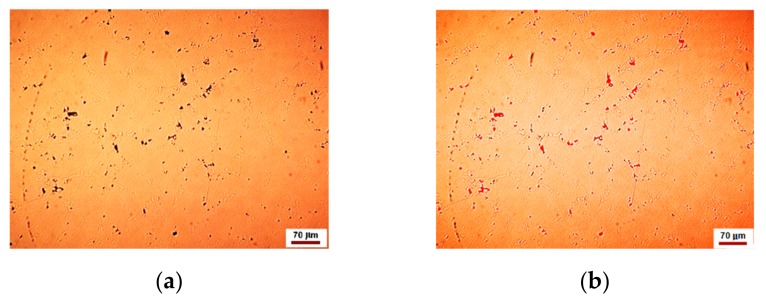
Inclusionary state associated with Figure 2a: (**a**) original microstructure image; (**b**) processed image; (**c**) histogram of the phase distribution; (**d**) histogram of the inclusion size distribution.

**Figure 6 materials-12-04199-f006:**
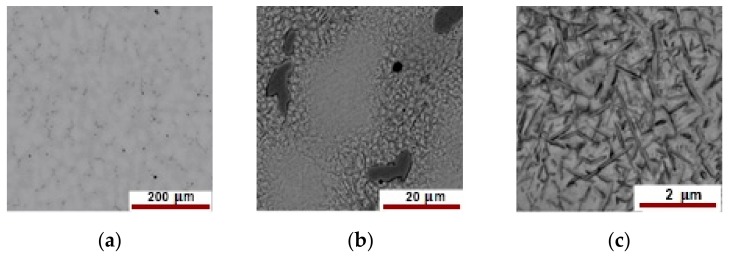
Representative SEM images of a eutectic area at different scales: (**a**) 200 µm; (**b**) 20 µm; (**c**) 5 µm.

**Figure 7 materials-12-04199-f007:**
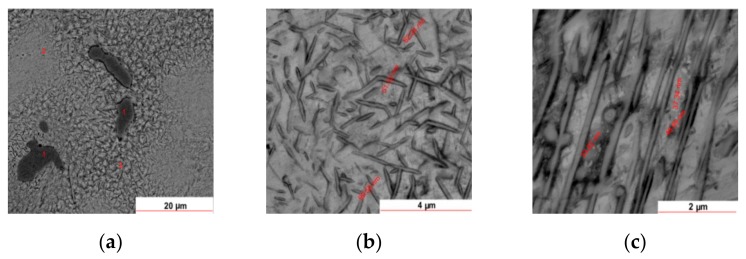
SEM images of another area of eutectic at different scales: (**a**) 20 µm; (**b**) 4 µm; (**c**) 2 µm.

**Figure 8 materials-12-04199-f008:**
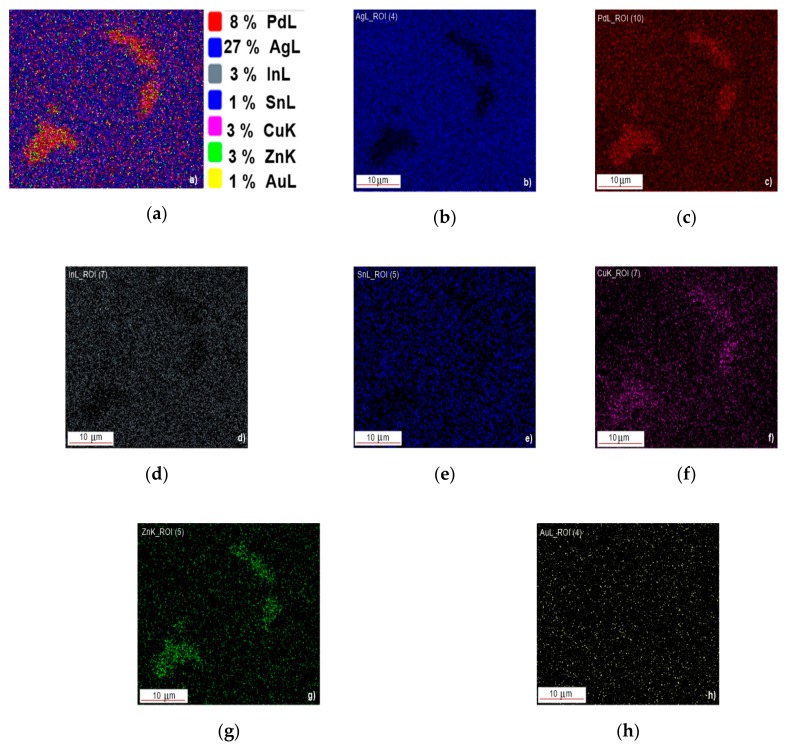
EDS maps of Ag, Pd, Cu, Zn, In, Sn, and Au elements associated with Figure 7a: (**a**) overlapped distributions of Pd, Ag, In, Sn, Cu, Zn, Au; (**b**) distribution of Ag; (**c**) distribution of Pd; (**d**) distribution of In; (**e**) distribution of Sn; (**f**) distribution of Cu, (**g**) distribution of Zn; (**h**) distribution of Au.

**Figure 9 materials-12-04199-f009:**
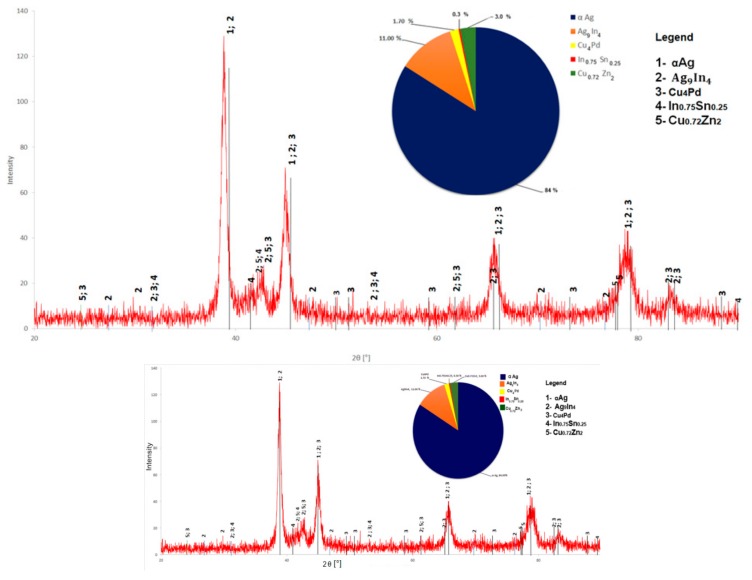
XRD pattern of the new alloy.

**Figure 10 materials-12-04199-f010:**
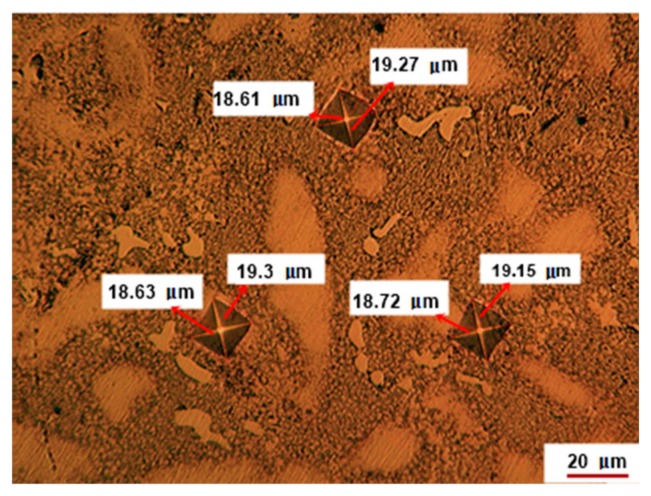
Optical images of the indentation traces with Vickers stylus.

**Figure 11 materials-12-04199-f011:**
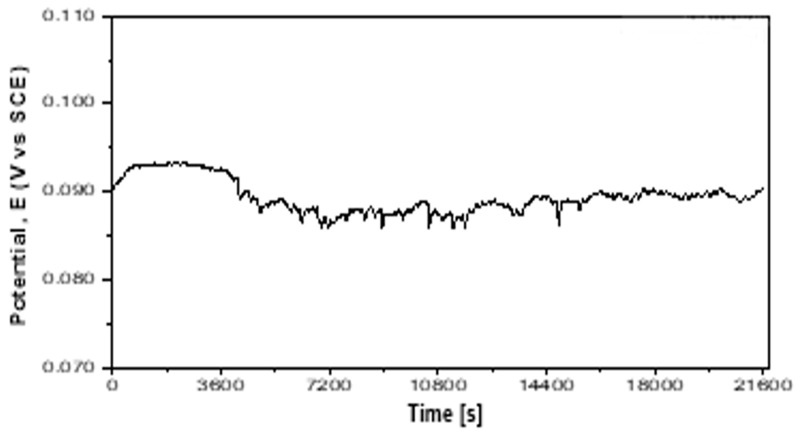
The E_OC_ evolution vs. time.

**Figure 12 materials-12-04199-f012:**
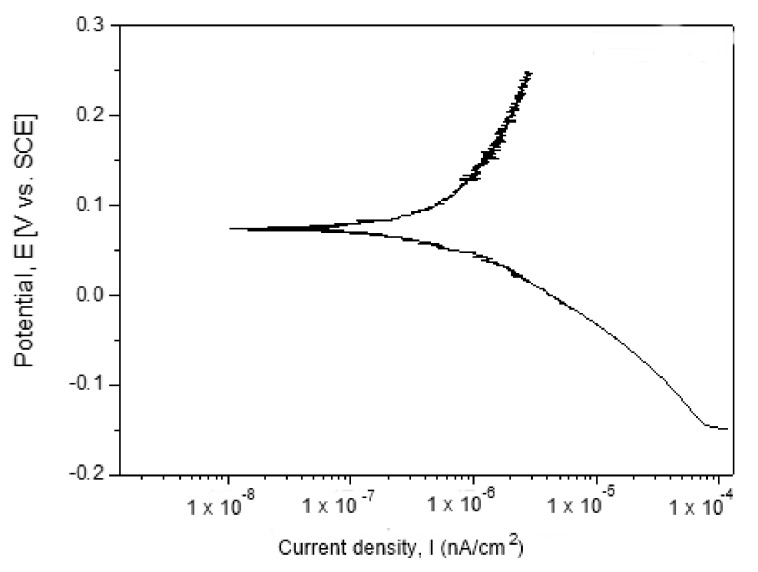
Tafel plots of the new alloy.

**Figure 13 materials-12-04199-f013:**
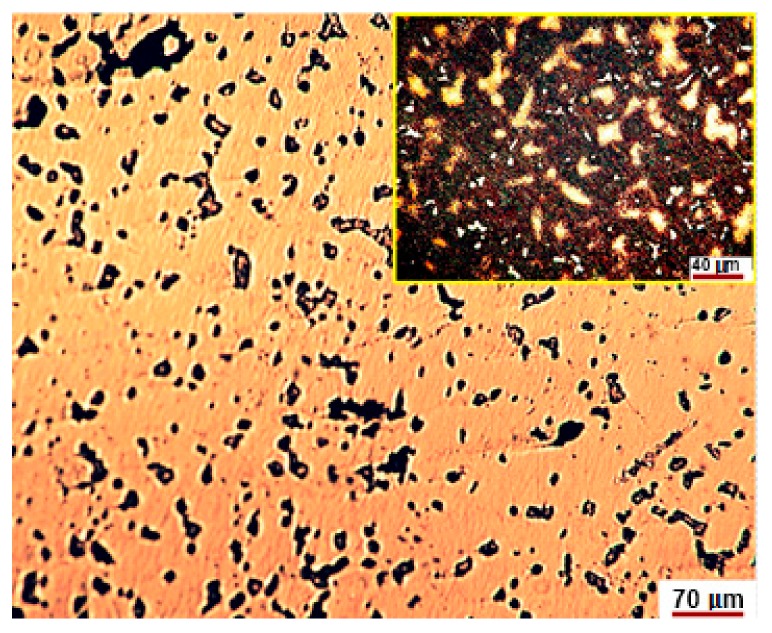
OM image of the corroded surface and of the blank (inset image) surface.

**Table 1 materials-12-04199-t001:** Elemental compositions of the reference alloy, of the designed one and the measured composition [wt%].

Element/Alloy	Ag	Au	Pd	Zn	In	Sn	Cu
Designed	58.0	2.0	24.0	2.0	1.5	1.5	11.0
as cast alloy	57.72	1.94	24.4	1.74	1.42	1.17	11.74
U (95%)	0.07	0.03	0.06	0.06	0.03	0.04	0.08

**Table 2 materials-12-04199-t002:** Chemical compositions of the spotted areas onto Figure 7a.

	Zone 1	Zone 1	Zone 2	Zone 2	Zone 3	Zone 3
Element/Alloy	Weight %	Atomic %	Weight %	Atomic %	Weight %	Atomic %
Pd L	22.9	21.14	25.62	24.92	23.3	21.52
Ag L	58.73	53.47	60.93	58.48	59.51	54.23
In L	0.01	0	1.69	1.52	1.14	0.98
Sn L	0.16	0.13	1.93	1.68	2.09	1.73
Cu K	13.19	20.39	7.46	12.15	12.44	19.25
Zn K	2.35	3.53	-	-	1.52	2.29
Au L	2.67	1.33	2.38	1.25	-	-

**Table 3 materials-12-04199-t003:** The values of the measurands of the electrochemical corrosion test carried on the new alloy.

E_oc_ (mV)	E_cor_ (mV)	I_corr_ (nA/cm^2^)	b_c_ (mV)	b_a_ (mV)	Rp (kΩ × cm^2^)	CR (µm/an)
+90.47	+74.73	916	108.31	380.74	40.02	23.96

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
