# Peer review of "Investigation of the Microstructure, Hardness and Corrosion Resistance of a New 58Ag24Pd11Cu2Au2Zn1.5In1.5Sn Dental Alloy"

_materials, 2019, doi:10.3390/ma12244199_

Round 1

Reviewer 1 Report

There should be some direct results to prove the idea of Fig8

Author Response

Thank you for your review of our paper. The authors greatly appreciate your comments and suggestions. We have answered each of your points below:

Crt.No.

Reviewer’s comment

Response

1.

Moderate English changes required 

The English of the entire manuscript was revised by a qualified person as is proven in the attached file.

2.

There should be some direct results to prove the idea of Fig. 8

The microstructure features shown in Fig. 7  are correlated with element distributions in Fig. 8. Thus, the lacking of Ag and increasing occurrence of Zn in the lower left corner  of Fig. 8 can be assigned to the occurrence of a compound in the same area in Fig. 7.a

Reviewer 2 Report

Present paper deals with investigation microstructure, hardness and corrosion resistance of a newly developed low-noble dental alloy. The topic is actual and fits the journal’s scientific area. Proper investigation methods are used. All references are cited in the text and there are not missing references in the list.

The manuscript contains some technical and grammatical errors that should be corrected.

English level: the whole manuscript should be checked and the right terms should be used. Title: the title does not correspond to the manuscript content. It is better to use “investigation of the microstructure, hardness and corrosion resistance” instead of “complex characterization”. Abstract: it is informative enough Some wrong terms, marked in yellow, are used (“higher-noble” is the right term). The last sentence shows not novelty. The complete characterization of a metal alloy includes all these methods. So, the sentence: “The microstructural features, the inclusionary state and the phase natures into 28 58Ag24Pd11Cu2Au2Zn1.5In1.5Sn revealed by consecrated methods are the other novelties 29 presented in the paper” should be omitted. Introduction: row 75: the statement “…. the way in which it was quantified the microstructure parameters of this alloy based on 75 computer image analysis is a novelty in the field.” does not show any novelty. This procedure is often used for evaluation of the microstructure. Materials and methods: row 105: the whole sentence, marked in yellow, is unnecessary. The specialists in the area know the application of optical microscopy. Results and Discussion: row 160: there is some misunderstanding with the figures’ numbering: “The The inclusionary state of the area in 1.a is shown in Figure 5.”, But in the caption of Fig. 5 is written: “Figure 5. Inclusionary state associated to Fig. 2a: a) original microstructure image…..”. Please check and correct. Row 264: there are grammatical errors in the sentence: “The studied alloy consists mainly of an eutectic (brown areas in Figs. 1-3) that engulfed Ag-Cu, 264 Cu-Zn, Ag-Cu, Ag-In, In-Sn binary compounds and, the most probably, ternary ones of the form Pd-265 Cu-Zn (Fig. 8) that nucleate and growth before eutectic solidification of the matrix”. Conclusions: they do not correspond to the manuscript content and should be basically revised. The main finding should be put in the conclusions. Row 276: only hardness is investigated in this paper and not all mechanical properties. Rows 283-285: the whole sentence, marked in yellow, is not suitable for conclusion. Rows 290-293: Using all mentioned methods this is not novelty, but standard characterization procedure, So, it should be omitted.

Author Response

Thank you for your review of our paper. The authors greatly appreciate your comments and suggestions.  Our answers to your points are as follows:

Crt.No.

Reviewer’s comment

Response

1.

English level: the whole manuscript should be checked and the right terms should be used.

The English of the entire manuscript was revised by a qualified person as is proven in the attached file.

2.

Title: the title does not correspond to the manuscript content. It is better to use “investigation of the microstructure, hardness and corrosion resistance” instead of “complex characterization”.

The title was changed i.e. “Investigation of the microstructure, hardness and corrosion resistance of a New 58Ag24Pd11Cu2Au2Zn1.5In1.5Sn Dental Alloy”

3

Abstract: it is informative enough Some wrong terms, marked in yellow, are used (“higher-noble” is the right term).

Higher-carat and lower-carat terms were replaced by higher-noble and lower-noble, respectively

4

The last sentence shows not novelty.  The complete characterization of a metal alloy includes all these methods.

So, the sentence: “The microstructural features, the inclusionary state and the phase natures into 28 58Ag24Pd11Cu2Au2Zn1.5In1.5Sn revealed by consecrated methods are the other novelties

presented in the paper” should be omitted.

The sentence was omitted

5

 Introduction: row 75: the statement “…. the way in which it was quantified the microstructure parameters of this alloy based on  computer image analysis is a novelty in the field.” does not show any novelty. This procedure is often used for evaluation of the microstructure

The sentence was omitted

6

Materials and methods: row 105: the whole sentence, marked in yellow, is unnecessary. The specialists in the area know the application of optical microscopy.

The sentence was omitted

7

 Results and Discussion: row 160: there is some misunderstanding with the figures’ numbering: The The inclusionary state of the area in 1.a is shown in Figure 5.”, But in the caption of Fig. 5 is written: “Figure 5. Inclusionary state associated to Fig. 2a: a) original microstructure image…..”. Please check and correct.

The numbering was corrected

8

Row 264: there are grammatical errors in the sentence: “The studied alloy consists mainly of an eutectic (brown areas in Figs. 1-3) that engulfed Ag-Cu, 264 Cu-Zn, Ag-Cu, Ag-In, In-Sn binary compounds and, the most probably, ternary ones of the form Pd-265 Cu-Zn (Fig. 8) that nucleate and growth before eutectic solidification of the matrix”. 

The sentence was corrected. Rows 273-276

9

Conclusions: they do not correspond to the manuscript content and should be basically revised.  The main finding should be put in the conclusions.

The conclusions were revised (see the attached revised manusctipt)

10

Row 276: only hardness is investigated in this paper and not all mechanical properties. Rows 283-285: the whole sentence, marked in yellow, is not suitable for conclusion.

Rows 283-285 were omitted

11

            Rows 290-293: Using all mentioned methods this is not novelty, but standard characterization procedure, So, it should be omitted. 

Rows 290-293 were omitted

Reviewer 3 Report

There is some missunderstanding.

The wrong manuscript is given for review.

The manuscript with title: "Effect of ZrC Formation on Microstructure Evolution 3 of the Carbon Phase in Polymer Derived ZrC-C4"

is presented instead of: "Complex Characterization of a New 58Ag24Pd11Cu2Au2Zn1.5In1.5Sn Dental Allo"

I appologize, but I can not do the review untill I receive the right manuscript.

Author Response

It is not the case to answer!

Round 2

Reviewer 3 Report

The manuscript is edited according to the reviewer's comments. Only final check of English gramar and spelling is needed.